# Mangiferin Represses Inflammation in Macrophages Under a Hyperglycemic Environment Through Nrf2 Signaling

**DOI:** 10.3390/ijms252011197

**Published:** 2024-10-18

**Authors:** Ravichandran Jayasuriya, Kumar Ganesan, Kunka Mohanram Ramkumar

**Affiliations:** 1Department of Biotechnology, School of Bioengineering, SRM Institute of Science and Technology, Kattankulathur 603 203, Tamil Nadu, India; jayasuriya199@gmail.com; 2School of Chinese Medicine, Li Ka Shing Faculty of Medicine, The University of Hong Kong, 10 Sassoon Road, Pokfulam, Hong Kong 999077, China; kumarg@hku.hk

**Keywords:** macrophages, mangiferin, Nrf2 activation, inflammation, antioxidant response

## Abstract

Inflammation in macrophages is exacerbated under hyperglycemic conditions, contributing to chronic inflammation and impaired wound healing in diabetes. This study investigates the potential of mangiferin, a natural polyphenol, to alleviate this inflammatory response by targeting a redox-sensitive transcription factor, nuclear factor erythroid 2-related factor 2 (Nrf2). Mangiferin, a known Nrf2 activator, was evaluated for its ability to counteract the hyperglycemia-induced inhibition of Nrf2 and enhance antioxidant defenses. The protective effects of mangiferin on macrophages in a hyperglycemic environment were assessed by examining the expression of Nrf2, NF-κB, NLRP3, HO-1, CAT, COX-2, IL-6, and IL-10 through gene and protein expression analyses using qPCR and immunoblotting, respectively. The mangiferin-mediated nuclear translocation of Nrf2 was evidenced, leading to a robust antioxidant response in macrophages exposed to a hyperglycemic microenvironment. This activation suppressed NF-κB signaling, reducing the expression of pro-inflammatory mediators such as COX-2 and IL-6. Additionally, mangiferin decreased NLRP3 inflammasome activation and reactive oxygen species accumulation in hyperglycemia exposed macrophages. Our findings revealed that mangiferin alleviated hyperglycemia-induced reductions in AKT phosphorylation, highlighting its potential role in modulating key signaling pathways. Furthermore, mangiferin significantly enhanced the invasiveness and migration of macrophages in a hyperglycemic environment, indicating its potential to improve wound healing. In conclusion, this study suggests that mangiferin may offer a promising therapeutic approach for managing inflammation and promoting wound healing in diabetic patients by regulating Nrf2 activity in hyperglycemia-induced macrophages.

## 1. Introduction

Macrophages are crucial players in the early inflammatory phase of wound healing, which usually occurs on the first day after homeostasis is restored [1]. These cells are programmed to perform in a sequential manner within the wound environment, with inflammatory macrophages arriving first to impart an inflammatory milieu to elicit invading pathogens, followed by anti-inflammatory macrophages to dampen inflammation, followed by tissue remodeling macrophages that contribute to wound repair. This cascade of macrophage activities is essential for effective wound healing [2]. However, in diabetic wounds, this process is either absent or significantly impaired, leading to chronic inflammation and delayed wound healing [1].

One significant comorbidity in diabetic patients is the development of diabetic foot ulcers (DFUs), which are characterized by an inability of the wounds to heal naturally and in a timely manner [3]. Elevated glycemic levels and stress-related factors in diabetic subjects disrupt the tightly regulated programming of macrophages [4]. In diabetic wounds, this disruption results in a persistent inflammatory state, driven by the continuous recruitment of pro-inflammatory macrophages (M1) to the wound site. As a result, the insufficient release of anti-inflammatory cytokines impairs the wound healing process [5].

M1 macrophages, characterized by a pro-inflammatory phenotype, are classically activated by IFN-γ, a cytokine that is significantly elevated in diabetic conditions. These macrophages release pro-inflammatory cytokines and reactive oxygen species (ROS), worsening tissue damage and prolonging inflammation. In contrast, M2 macrophages, which represent the alternative phenotype, have anti-inflammatory and tissue-repairing properties. Activated by IL-4 and IL-13, M2 macrophages are essential for resolving inflammation, promoting wound healing, and maintaining tissue homeostasis [6]. However, in diabetic wounds, this balance is disrupted, leading to a dominance of M1 macrophages and an impaired transition to the M2 phenotype. A recent study by Huang et al. demonstrated the activation of M1 macrophages in response to stimulation with IFN-γ and LPS. Furthermore, the study highlighted that hyperglycemia exacerbates M1 polarization in diabetic mice, further intensifying the inflammatory state and hindering proper wound healing [7]. This imbalance between M1 and M2 macrophages is critical in the pathogenesis of chronic inflammation in diabetes [8]. The failure to properly regulate macrophage polarization contributes to the inability of the wound to transition from the inflammatory to the tissue-remodeling phase, which is necessary for healing. Therefore, controlling inflammation is essential to prevent the accumulation of macrophages in a pro-inflammatory state and to ensure an effective transition to the subsequent anti-inflammatory and tissue-repair phases. Restoring the balance between M1 and M2 macrophages, possibly by promoting M2 polarization, presents a potential therapeutic strategy to counteract excessive inflammation and improve wound healing in diabetic patients.

Hyperglycemia not only impairs macrophage function but also affects neutrophils by reducing their phagocytic activity and, more importantly, increasing the formation of neutrophil extracellular traps (NETs). This process triggers stress and inflammation, leading to further damage to surrounding cells and tissues [9]. Additionally, hyperglycemia exacerbates the proliferation of CD4+ T cells by activating the MAPK pathway [10]. These conditions, combined with the excessive release of free radicals, induce a pro-inflammatory state that weakens immune surveillance. Therefore, there is a need for therapies that reduce inflammation and restore normal immune function, including the regulation of macrophages.

Nuclear factor erythroid 2-related factor 2 (Nrf2), a redox sensitive transcription factor is tightly associated with inflammation and is proven for its role in mediating protection against xenobiotic stresses under various stimuli. Under normal conditions, Nrf2 binds to Keap1, leading to its ubiquitination and subsequent degradation by the proteasome. This ubiquitination and proteasomal degradation are halted when triggered by xenobiotic stresses, resulting in the nuclear translocation of Nrf2, followed by the transcription of antioxidant genes [11]. Nrf2 plays a key role in ameliorating oxidative stress induced under various pathological stimulus including the excessive synthesis of ROS. Hyperglycemic conditions, as seen in diabetic subjects, trigger mitochondrial and endoplasmic stress through the activation of advanced glycation end products (AGEs). This in turn stimulates the excessive accumulation of ROS, promoting cellular damage [12], which are neutralized by the activation of Nrf2. Along with this, Albert-Garay at al. reported an increase in oxidative milieu decreases Nrf2 signaling in hyperglycemia-exposed rat Müller retinal cells [13]. Excessive ROS accumulation aggravates oxidative stress and mitochondrial dysfunction, which impairs polarization shift from M1 to M2, thereby maintaining a pro-inflammatory environment, and along with this, impaired phagocytosis leads to the accumulation of debris and necrotic cells, exacerbating tissue damage [14].

The role of Nrf2 in modulating inflammatory responses by maintaining a regulated crosstalk with another transcription factor, NF-κB (nuclear factor kappa-light-chain-enhancer of activated B cells) [15]. The NF-κB is an inducible transcription factor that becomes activated based on the inhibition of its partner, IκB (inhibitor of nuclear factor kappa B), is degraded. This aids the translocation of NF-κB to nucleus, where it promotes the transcription of pro-inflammatory genes [16]. The Nrf2/NF-κB system functions in a highly coordinated manner, where decreased antioxidant defense induces inflammation through nuclear translocation of NF-κB, and vice versa. NF-κB activation can, in turn, suppress Nrf2 activity, creating a feedback loop that influences the balance between oxidative stress and inflammation [17,18]. This skeptical regulation of Nrf2 made as an attractive drug target in the management of various diseases including diabetes [19,20].

Hyperglycemia also affects multiple pathways other than NF-κB, including JAK/STAT, PKC, AGE, MAPK, and TLR signaling, which together lead to the over-production of pro-inflammatory cytokines like IL-6, IL-1β, and TNF-α, ultimately leading to inflammation and tissue damage [21]. While these pathways are being influenced by hyperglycemia, NF-κB plays a main role as a master regulator of inflammation by controlling the transcription of inflammatory genes as well as the assembly of inflammasome complexes, which further upregulates inflammation to neighboring cells [22]. On the other hand, Nrf2 has always been recognized as a potent antioxidant-regulation transcription factor, which helps reduce inflammation by suppressing NF-κB levels [23,24,25]. Thus, the current study focuses on Nrf2/NF-kB system influenced by hyperglycemia in macrophages.

An array of small molecules disrupts the Nrf2-Keap1 complex through electrophilic dissociation, promoting antioxidant system through Nrf2 nuclear translocation. For example, the pharmacological activation of Nrf2 by small molecules like naringenin [26], morin [27], and pterostilbene [28] has been proven to ameliorate diabetes-induced effects in experimental models. In addition, a recent study by Ganesh and Ramkumar highlighted the wound healing potential of pterostilbene, including ability to promote the transition to anti-inflammatory macrophages in experimental diabetic wounds [29]. Furthermore, several clinical trials have been conducted, and, notably with Bafiartam^®^ and Vemurity^®^, which contain monomethyl fumarate (MMF) and diroximel fumarate (DMF), respectively, as active compounds that mimic the molecular structure of Nrf2. Both drugs have been approved by FDA for the treatment of multiple sclerosis [30,31]. These compounds disrupt the Keap1-Nrf2 interaction and promote Nrf2 translocation, thereby enhancing antioxidant and anti-inflammatory effects to counteract the progression of multiple sclerosis. IT should be noted that 0.1% of RTA 408, a formulation from Reata Pharmaceuticals, was established to enhance wound healing capacity in diabetic mice, targeting Nrf2 signaling [32]. Nrf2 activation can be triggered by the phosphorylation of upstream kinases like ERK, MAPK, and AKT. In particular, the PI3K/AKT pathway plays a critical role in regulating Nrf2-ARE signaling, and promotes protection against various stresses under hyperglycemia [33]. Subsequently, the ARE-Nrf2 system allows the transcription of HO-1 that offers protection to mitochondria in SH-SY5Y cells exposed to hydrogen peroxide [34].

In this study, we aimed to investigate the anti-inflammatory activity of mangiferin in macrophages under hyperglycemic stimulus. While we previously established the Nrf2-regulating activity of mangiferin in endothelial cells under hyperglycemic stress [35], the current study focuses on the Nrf2 activation potential of mangiferin in macrophages. Specifically, we examined how Nrf2 activation by mangiferin influences the inflammatory response induced by hyperglycemic stress in these immune cells. We utilized PMA to induce macrophage differentiation in monocytes, which were then exposed to a hyperglycemic environment to simulate diabetic wounds. These PMA-induced macrophages exhibit a pro-inflammatory phenotype, making them an ideal model for studying inflammatory disorders [36,37]. Our diabetic model integrates the use of elevated glucose levels with a cytokine cocktail, including the tumor necrosis factor (TNF)-α and interferon (IFN)-γ, which are predominantly found in the wound fluids or in the plasma of DFU patients [38,39]. The inclusion of TNF-α is particularly crucial, as its dysregulation can severely impair the inflammatory phase of wound healing and impede wound closure [40]. Therefore, this model provides a robust platform for evaluating the efficacy of mangiferin in modulating inflammation and promoting wound healing.

## 2. Results

### 2.1. PMA Promotes Differentiation of THP1 Monocytes to Macrophages

THP1 monocytes were seeded in a 96-well plate and treated to various concentrations of PMA to determine cell viability. The viability percentage of cells decreased significantly at 100 ng/mL of PMA (Figure 1a); thus this concentration was used in subsequent experiments. Macrophage differentiation was confirmed by examining the expression of CD18 and CD14, which are macrophage and monocyte markers, respectively. Figure 1b shows a significant increase (*p* < 0.001) in the expression of CD18, while the expression of CD14 decreased by nearly 10-fold (Figure 1c), demonstrating successful differentiation. Figure 1d represents the fully differentiated macrophages from THP1 monocytes following PMA treatment (100 ng/mL) and a 24 h recovery period. These macrophages were used to investigate the efficacy of mangiferin in a HGM.

### 2.2. Mangiferin Mediated Nrf2-Activation in Macrophages

Firstly, the viability of macrophages on exposure to mangiferin was estimated using Alamar blue assay. Figure 2a shows reduced viability at concentrations above 25 μM, leading us to use up to 25 μM for further studies. As depicted in Figure 2b, we found a dose-dependent increase in the expression of Nrf2 mRNA at 10 and 25 μM concentration. Additionally, to investigate the role of mangiferin on Nrf2 activation in macrophages, we treated the macrophages with mangiferin (5, 10, and 25 μM), and extracted the nuclear and cytosolic fractions. From this, we observed a significant upregulation in nuclear Nrf2 and downregulation of cytosolic Nrf2, emphasizing the nuclear translocation (Figure 2c).

### 2.3. Mangiferin Protects Hyperglycemia-Induced Macrophages through Upregulation of Nrf2

To assess the cytoprotective effects of mangiferin, we measured macrophage viability after treatment with various concentrations of mangiferin. As shown in Figure 3a, there was a dose-dependent increase in macrophage viability with significant improvements at 10 μM (*p* < 0.05) and 25 μM (*p* < 0.01) under hyperglycemic microenvironments (HGMs). Based on these results, we selected these concentrations to further investigate Nrf2 activation and the inflammatory response. To evaluate the effect of mangiferin on Nrf2 and its downstream targets, we exposed macrophages to 10 and 25 μM of mangiferin. Both the protein (Figure 3b) and gene (Figure 3c) expression of Nrf2 were significantly reduced (*p* < 0.001) under HGM but were progressively increased with mangiferin treatment at 10 μM (*p* < 0.05) and 25 μM (*p* < 0.01).

### 2.4. Effect of Mangiferin on Upstream and Downstream Targets of Nrf2 in Macrophages under a Hyperglycemia Microenvironment

The translocation and activation of Nrf2 involve phosphorylation by upstream kinases, including AKT. Therefore, we investigated the effect of mangiferin on AKT phosphorylation in macrophages exposed to hyperglycemic microenvironments (HGMs). As shown in Figure 4a, macrophages treated with mangiferin exhibited a significant increase in AKT phosphorylation. Conversely, AKT phosphorylation was significantly decreased under hyperglycemic conditions (*p* < 0.001). This reduction in phosphorylation was reversed with mangiferin treatment, resulting in a significant dose-dependent increase at both 10 and 25 μM.

Additionally, we assessed the mRNA expression levels of Nrf2 downstream targets, including heme oxygenase 1 (HO-1) and catalase (CAT). As demonstrated in Figure 4b,c, HO-1 and CAT expression levels were significantly decreased under hyperglycemic conditions (*p* < 0.001), but were significantly upregulated with mangiferin treatment (*p* < 0.01).

### 2.5. Mangiferin Modulates the Inflammatory Response Induced by Hyperglycemic Microenvironment in Macrophages

To investigate the role of mangiferin in regulating NF-κB expression, we utilized HEK293 cells expressing the NF-κB luciferase reporter system. The reporter cells were exposed to HGM for 24 h while simultaneously co-treated with a high dose of mangiferin (25 μM) at varying time points. After treatment, the cells were lysed, and luciferase activity was measured to assess NF-κB activity (Figure 5a). After 24 h of exposure to HGM, we observed nearly a seven-fold increase in the NF-κB luciferase signal. However, this increase was significantly downregulated after 12 h of co-treatment with mangiferin (*p* < 0.05). By 24 h of co-treatment, the luciferase signal had decreased by almost 10-fold (*p* < 0.01) compared to cells exposed to HGM alone. Notably, at this 24 h mark, NF-κB expression levels were nearly equivalent to those of the control group, indicating effective suppression of the inflammatory response. Furthermore, Nrf2 activation was also upregulated at this time point, as observed in previous sections, likely contributing to the repression of NF-κB activation under hyperglycemic conditions. Therefore, 24 h of mangiferin treatment was selected for further experiments, as it effectively modulates both NF-κB and Nrf2 signaling pathways.

Further, the effect of mangiferin on HGM-exposed macrophages were studied in order to ascertain its role on inflammation regulation in macrophages. Figure 5b shows a more than 2.5-fold upregulation (*p* < 0.001) in the expression of NF-kB protein, which was reduced after treatment with mangiferin at 10 (*p* < 0.05) and 25 μM (*p* < 0.01) concentrations. This NF-kB activation led to a significant increase (1.8-fold) in the pro-inflammatory marker cyclooxygenase (COX-2) (Figure 6a). Mangiferin treatment was observed to lessen the expression of COX-2 in a dose-dependent manner. In addition, the gene expression of a pro-inflammatory marker, IL-6 (Figure 6b), and an anti-inflammatory marker, IL-10 (Figure 6c) were assessed. The expression of IL-6 increased to 2.5-fold, with a decrease in IL-10 (2-fold), suggesting a heightened inflammatory response in regard to HGM exposure. Mangiferin treatment reversed this effect by reducing IL-6 and increasing IL-10 expression, thereby mitigating HGM-induced inflammation.

### 2.6. Mangiferin Attenuates Hyperglycemic Microenvironment-Induced Inflammasome Activation in Macrophages

As shown in Figure 7a, HGM induced a 2.5-fold increase in NLRP3 activation compared to the control, which was reduced by mangiferin treatment at 10 μM (*p* < 0.05) and 25 μM (*p* < 0.001). We then studied the IL-1β (Figure 7b) and IL-18 (Figure 7c) secretion in response to NLRP3 activation and mangiferin treatment. The mature/active forms of IL-1β and IL-18 were relatively high (*p* < 0.001) under HGM due to NLRP3 activation. Mangiferin treatment led to a dose-dependent decrease in IL-1β and IL-18 levels by inhibiting NLRP3 activation.

### 2.7. Mangiferin Ameliorates Intracellular Reactive Oxygen Species Formation in Macrophages Triggered by Hyperglycemic Microenvironment

The accumulation of intracellular reactive oxygen species (ROS) is a crucial mediator of NLRP3 activation, particularly in inflammation-driven cells such as macrophages, which forms a feedback loop with inflammasome activation [41]. To investigate, we used H_2_DCFDA probes to monitor ROS generation in live macrophages, observed using a fluorescent microscopy. Figure 8 shows increased ROS-induced fluorescence in macrophages exposed to HGM. Treatment with mangiferin at 10 and 25 μM reduced the intensity of green fluorescence, indicating decreased ROS production. The data showed a nearly two-fold increase in ROS production under HGM, which was mitigated by mangiferin treatment. Further, there was no significant difference in the cell count assessed using DAPI staining.

### 2.8. Effect of Mangiferin on Migration and Invasion of HGM-Induced Macrophages

The suppression of NLPR3 promotes anti-inflammatory phenotype in macrophages and thereby helps in the healing of diabetic wounds [42]. The mangiferin-mediated suppression of NLRP3 being established, the ability of macrophages to migrate and invade in a hyperglycemic environment was assessed using a Transwell chamber. HGM exposure resulted in significantly reduced migration (*p* < 0.01) and invasion (*p* < 0.001) compared to the control. However, treatment with 10 and 25 μM mangiferin significantly enhanced both migration and invasion, demonstrating improved macrophage function under HGM conditions (Figure 9).

## 3. Discussion

Diabetic wounds are characterized by prolonged inflammation, largely due to the altered programming of tissue-resident macrophages [43]. Chronic hyperglycemia and elevated oxidative stress impair the transition of macrophages from a pro-inflammatory to an anti-inflammatory phenotype. This leads to the sustained release of pro-inflammatory markers such as TNF-α, IL-6, and IL-1β [44], while the secretion of anti-inflammatory cytokines like IL-10 is diminished, resulting in tissue damage and a failure to progress to the proliferative phase. The imbalance between M1 macrophages, which promote inflammation, and M2 macrophages, which support tissue repair, plays a critical role in this process [45]. Promoting anti-inflammatory polarization could help restore balance and enhance wound healing in diabetic patients.

In addition, diabetic wounds are characterized by increased oxidative stress due to an imbalance between the production of reactive oxygen species (ROS) and the body’s ability to neutralize them via antioxidants [46]. Hyperglycemia in diabetic subjects contributes to excessive ROS generation, leading to cellular damage and further compromising the wound microenvironment. This reduced antioxidant status makes the wound microenvironment more hostile and compounding difficulties in effective wound healing [47]. Therefore, controlling both stress environment and inflammation is crucial to reduce tissue damage and restore the progression of the wound to the next phase of healing [48].

Conditions such as hyperglycemia, ROS and advanced glycation end-products (AGEs) trigger the activation of the master transcription factor of inflammation, NF-κB [49]. The nuclear translocation of NF-κB promotes the expression of pro-inflammatory cytokines, leading to a chronic inflammatory state that exacerbates oxidative damage, which ultimately impairs the tissue repair mechanism [50]. Enhancing the body’s antioxidant system can help to overcome this inflammatory and oxidative milieu, thereby supporting enhanced tissue repair [51]. Nrf2, a stress-activated transcription factor, helps in maintaining cellular homeostasis by regulating the expression of antioxidant genes. Nrf2 and NF-κB share a negative relationship, representing a critical axis in the regulation of inflammation and oxidative stress [15].

The activation of Nrf2 has shown promising results in treating various diseases, including diabetes and its complications. Upon activation, Nrf2 translocates to the nucleus, where it binds to the antioxidant response element (ARE) and promotes the expression of a wide array of antioxidant and cytoprotective genes, such as heme oxygenase-1 (HO-1), catalase (CAT), superoxide dismutase (SOD), NAD (P)H quinone dehydrogenase 1 (NQO1), glutathione S-transferase (GST), and glutathione peroxidase (GPx) [52]. These genes collectively reduce oxidative stress and modulate the inflammatory response, facilitating tissue repair.

Nrf2 activators have received increasing interest in the recent years regarding various diseases. Sulforaphane, for example, is a widely explored phenolic compound that primarily targets NF-κB to repress inflammation through Nrf2 activation [53,54]. Other natural compounds include curcumin, resveratrol, epigallocatechin gallate, bardoxolone methyl, dimethyl fumarate, and oltipraz, which have been shown to regulate Nrf2 and reduce inflammation in preclinical studies [55]. While resveratrol [56] and epigallocatechin gallate [57] have demonstrated pro-angiogenic effects, promoting the formation of new blood vessels and facilitating wound healing, curcumin also exhibits pro-apoptotic activity, specifically targeting the stalled inflammatory cells, which helps resolve the chronic inflammation that often delays wound healing [58]. By clearing these cells, curcumin supports the progression of the healing process beyond the inflammatory phase. In addition to pre-clinical studies, the activation of Nrf2 has proven to be highly useful in clinical settings as well. Tecfidera^®^, a dimethyl fumarate, is being recognized for its ability to induce Nrf2 and exert its anti-inflammatory effects in individuals with multiple sclerosis [59]. Formulations on MMF and DMF, such as Bafiartam^®^ and Vemurity^®^, are also used for the treatment of multiple sclerosis [30,31]. However, there is no direct evidence or active ongoing trials on Nrf2 activators for its application for wound healing purposes.

Diabetic wounds present significant challenges, often characterized by impaired healing due to oxidative stress and chronic inflammation. In this context, small molecules are increasingly studied for their antioxidant and anti-inflammatory potential, as they promote tissue repair, enhance angiogenesis, and modulate immune responses within the wound environment. In the present study, we investigated mangiferin, a natural phenolic xanthonoid, for its ability to activate the Nrf2 pathway and enhance its anti-inflammatory effects in macrophages exposed to a hyperglycemic environment. This approach specifically aims to mitigate the adverse inflammatory responses associated with hyperglycemia through utilizing mangiferin′s capacity to modulate inflammation via Nrf2 activation.

Mangiferin has a broad spectrum of beneficial effects that help mitigate various stress responses. It promotes antioxidant enzymes, facilitating recovery from CCl4-induced liver injury and inflammation through bile acid metabolism and the suppression of pro-fibrotic genes [60]. Additionally, it has been shown to improve insulin resistance via the p-AKT-mediated regulation of GLUT2 [61], which is crucial for increasing glucose uptake. Furthermore, mangiferin plays a vital role in maintaining energy homeostasis through AMPK phosphorylation [62]. Its potential antioxidant mechanisms have also demonstrated protective effects against Huntington’s disease-like symptoms in animal models [63]. Moreover, Zynamite^®^, a proprietary extract containing mangiferin, is consumed for its ability to improve cognitive function, reduce pain, enhance oxygenation, and alleviate muscle fatigue in humans [64,65].

A prior review by Victor et al. has highlighted various Nrf2 activators, including mangiferin, that accelerate wound healing in diabetic conditions. Although these activators have varied functions on promoting wound healing, they primarily focus on three key mechanisms, namely (i) dissociation from Keap1, (ii) activation by upstream kinases, and (iii) the prevention of Nrf2 ubiquitination [66]. While the existing studies have explored these mechanisms, our research focuses on the inflammation-mediated alterations of macrophages, induced in a diabetic environment. By examining this aspect, we aim to provide deeper insights into how mangiferin not only enhances Nrf2 signaling but also influences inflammatory response on macrophages, thereby offering a new therapeutic avenue for managing diabetic wounds.

Firstly, we evaluated the Nrf2 activation potential of mangiferin and observed a dose-dependent activation. Nrf2 is a nuclear protein, and hence we demonstrated Nrf2 nuclear translocation in macrophages on exposure to mangiferin. This confirmed mangiferin-mediated Nrf2 translocation, which would further lead to the transcription of antioxidant genes. We then examined the activity of Nrf2 in macrophages exposed to hyperglycemic stress and found that mangiferin significantly improved Nrf2 levels in these cells under hyperglycemic stress. This led to the increased expression of antioxidant genes such as HO-1 and CAT in macrophages under hyperglycemic stress. These data confirmed the mangiferin-mediated activation of Nrf2 and antioxidant defense in macrophages under hyperglycemic stress. In line with our observations in macrophages, Zhou et al. demonstrated that mangiferin inhibits Keap1, thereby promoting the translocation of Nrf2 to the nucleus and the subsequent transcription of antioxidant genes in SH-SY5Y cells [67]. Additionally, a study on neonatal rats highlighted the remarkable capacity of mangiferin to elevate Nrf2 expression by promoting the degradation of Keap1 [68].

As mentioned above, Nrf2 has a negative relationship with NF-κB, evidenced by our finding, where decreased Nrf2 levels under hyperglycemic stress increased the expression of NF-κB. This increase in NF-κB expression further increased the levels of its classical downstream COX-2 [69], as confirmed by protein expression analysis. This cascade resulted in an upregulation of the pro-inflammatory cytokine expression of IL-6 and a decrease in the expression of anti-inflammatory cytokine, IL-10. A recent report by Manríquez-Núñez et al. described the ability of macrophages to change a more pro-inflammatory phenotype under hyperglycemic conditions [70]. Moreover, Dissanayake et al. described the increased expression of pro-inflammatory cytokines in macrophages exposed to hyperglycemia [71]. In contrast, with increased Nrf2 expression induced by mangiferin in HGM-induced macrophages, we observed a dose-dependent reduction in the expression of NF-κB, COX-2, and IL-6. Furthermore, mangiferin also increased in the expression of IL-10 under hyperglycemic stress.

In diabetic wounds, excessive ROS production perpetuates the pro-inflammatory environment by further priming the action of NLRP3, driven in response to NF-κB activation or directly by oxidative stress environment [72]. This NLRP3 subsequently helps in the assembly of the inflammasome complex and the activation of caspase-1, which then cleaves pro-IL-1β and pro-IL-18 into their active forms, leading to their secretion in the wound site [73]. The secretion of mature IL-1β binds to its receptor, activating NF-κB signaling in other cells, thereby creating a feed-forward loop that sustains a prolonged inflammatory state in diabetic wounds. The assembly and functionality of NLRP3 can be compromised in the absence or reduced recruitment of NF-κB, which can be achieved by increasing antioxidant defense [74].

Given the crucial role of the NF-κB/NLRP3 system, Nrf2 activation is particularly beneficial as it modulates the regulation of and NLRP3, either through its interaction with NF-κB, or directly by combating oxidative stress. Notably, Lee et al. reported the increased expression of NLRP3 in the circulation of patients with type 2 diabetes [75]. In the current study, mangiferin-mediated Nrf2 activation, and subsequent inhibition of NF-κB implicates in reduced ROS accumulation in macrophages under hyperglycemic stress, as evidenced by DCFDA staining. This reduction in cellular ROS and NF-κB inhibits the NLRP3 activation, thus reducing IL-1β and IL-18 secretion. This cascade of events aided by mangiferin reduces the tissue being exposed to a highly inflammatory state and promotes the healing of the tissue.

The activation of NLRP3 inflammasome disrupts the ability of macrophages to polarize into an anti-inflammatory phenotype [76], which is essential for effective tissue repair. This disruption not only hampers their reparative functions but also impairs macrophage migration and invasion into areas requiring healing [77]. Macrophage migration and invasion are fundamental to ensure the timely clearance of pathogens, the resolution of inflammation, and the initiation of tissue repair. When these processes are impaired, the immune response is weakened, resulting in delayed wound healing and poor progression of the wound towards healing [78].

A recent study demonstrated that MCC950, an NLRP3 inhibitor, significantly promoted wound healing in a streptozotocin-induced diabetic rat model by inhibiting NLRP3 activation [79]. Additionally, in vitro findings showed that MCC950 improved migration of HUVECs in a scratch assay, which was decreased due to AGE-induced NLRP3 activation [79]. This highlights a clear inverse relationship between NLRP3 activation and cell migration, emphasizing the role of NLRP3 inhibition in promoting cell movement and wound healing. Consistent with these findings, our current study demonstrated that mangiferin improved macrophage migration and invasion under hyperglycemic conditions by suppressing NLRP3 activation. Mangiferin effectively restored macrophage function, allowing them to perform their essential role in cell migration and invasion, further supporting its potential as a therapeutic agent for improving wound healing in diabetic conditions. Altogether, the current study proved mangiferin improved antioxidant response through Nrf2 regulation, inhibiting hyperglycemic stress-induced inflammation in macrophages. An overall schematic representing the mechanism of action is represented as Figure 10.

## 4. Materials and Methods

### 4.1. THP1 Cell Culture and Macrophage Induction

THP1 monocytes were cultured in DMEM (Sigma, St. Louis, MO, USA) with 10% fetal bovine serum (Gibco, Waltham, MA, USA) and maintained in a humidified incubator with 5% CO_2_. THP1 monocytes were differentiated into macrophages by exposing the cells to phorbol 12-myristate 13-acetate (PMA, Abcam, Boston, MA, USA) for 24 h, followed by an additional 24-h incubation in DMEM. The duration for macrophage differentiation from THP1 monocytes was selected based on a previous study [80], while the concentration of PMA used in this study was determined based on our observations of surface marker expression (outlined in Section 2.1). These PMA-induced macrophages (referred to as macrophages hereafter) were then subjected to a hyperglycemic microenvironment (HGM, 33.3 mM glucose, and a cytokine cocktail of 20 ng/mL TNF-α and IFN-γ) for 72 h. The concentration of HGM was chosen to mimic the microenvironment observed in diabetic wounds [38,81]. TNF-α and IFN-γ as a cocktail sufficiently simulates M1 macrophage activation in context of diabetic wound models [38]. The macrophages were treated with mangiferin at various concentrations for 24 h. Mangiferin (MG, ≥98% purity; Sigma, USA) was dissolved in 0.01% (*v*/*v*) DMSO. The control group was exposed to the same volume of solvent (vehicle) to precisely assess the specific effects of mangiferin on macrophages. Importantly, previous studies have reported that DMSO concentrations up to 2.5% are non-toxic to RAW264.7 macrophages [82].

### 4.2. Cell Proliferation Assay

Macrophages were seeded onto a 96-well plate (20,000 cells/well) and exposed to different concentrations of mangiferin for 24 h. After this, cells were incubated with 0.1% Resazurin solution (Alamar Blue, Himedia, Mumbai, India) for an additional 4 h. Absorbance was measured at 570 and 600 nm using a microplate reader.

### 4.3. RNA Isolation and Real-Time PCR

Macrophages were harvested using Accutase solution (Sigma, USA), washed and subjected to RNA isolation with RNA isoplus/TRIzol reagent (Takara, Kyoto, Japan). Briefly, about 300 μL TRIzol was added to the cell pellet, placed on ice and lysed by vortexing for 30 min. The homogenate was added with 200 μL of chloroform, mixed by pipette and placed on ice for another 30 min with intermittent mixing. The mixture was then centrifuged at 12,500 rpm, and the clear supernatant was added with an equal volume of ice-cold isopropanol and incubated for 3 h. After centrifugation at 12,500 rpm, the pellet was washed with 70% ethanol, air-dried in a biosafety cabinet and dissolved in nuclease-free water (Bio-Basic, Markham, ON, Canada). The isolated RNA was quantified using a nanoquant spectrophotometer (Tecan, Männedorf, Switzerland) and reverse transcribed into cDNA using the PrimeScript RT reagent Kit (Takara, Japan) as per the manufacturer’s protocol. qRT-PCR was performed on a CFX-Connect Real-Time PCR (Bio-Rad, Hercules, CA, USA) using TB II SyBr green master mix (Takara, Japan). The forward and reverse primers for target genes were designed based on their mRNA sequences from the NCBI database using primer3plus software. Glyceraldehyde 3-phosphate dehydrogenase (GAPDH) was used as an internal control for normalization.

### 4.4. Whole Cell Protein Extraction

After experimental conditions, the macrophages were washed using PBS, harvested using accutase (Sigma, USA) and lysed using RIPA buffer (Sigma, USA) with protease inhibitor cocktail (Abcam, USA) as per the manufacturer′s instructions to extract whole-cell proteins. The contents were lysed by repeated vortexing, and maintained on ice for about 30 min. After homogenization, the contents were centrifuged at 12,500 rpm for 15 min and the supernatant was recovered to a new eppendorf tube. β-actin was used as internal control for normalization.

### 4.5. Nuclear and Cytoplasmic Fractionation

The effect of mangiferin on the nuclear translocation of Nrf2 was studied in macrophages by extracting the nuclear and cytoplasmic fractions using a commercially available kit (#ab113474, Abcam, USA), as per the manufacturer′s instructions. The harvested cells were washed with PBS, and the cytoplasmic fraction was separated first using pre-extraction buffer provided with the kit. Following this, the pellet was washed with PBS, and the nuclear protein was fractionated using the extraction buffer provided with the kit. Lamin-B1 and β-actin were used as internal controls for the nuclear and cytosolic fractions, respectively.

### 4.6. Immunobloting

The concentration of protein collected from both extraction methods was determined using Bradford′s reagent (Bio-Rad, USA). A normalized concentration of protein (40 μg) was electrophoresed on an SDS-PAGE gel and transferred to a nitrocellulose (NC) membrane using a semi-dry transfer unit (Bio-Rad, USA). The NC membrane was blocked using 3% BSA and probed with primary antibodies against Nrf2 (#ab92946, Abcam, USA), t-AKT (#sc-8312, Santa Cruz, Dallas, TX, USA), p-AKT-Ser 473 (#sc-7985, Santa Cruz, CA, USA), NF-κB-p65 (#sc-8008, Santa Cruz, CA, USA), COX-2 (#sc-19999, Santa Cruz, CA, USA), and NLRP3 (#MAB7578, R&D Systems, Minneapolis, MN, USA). After overnight incubation with primary antibodies, blots were washed, incubated with HRP-conjugated secondary antibody (Santa Cruz, USA) for 2 h, and analyzed using an enhanced chemiluminescence (ECL) kit (Bio-Rad, USA) on a Gbox chemi-documentation system (Syngene, Cambridge, UK). Membranes were then stripped (#ab282569, Abcam, USA) and re-probed with β-actin (#sc-47,778, Santa Cruz, USA) or Lamin B1 (#ab229,025, Abcam, USA) antibodies overnight, followed by incubation with secondary antibodies and imaging as described.

### 4.7. Enzyme-Linked Immunosorbent Assay

IL-1β and IL-18 levels in the supernatant of cultured macrophages under HGM, with or without mangiferin, were measured using the Elikine Human IL-1β ELISA kit (#KTE6013, Abbkine, Atlanta, GA, USA) and the Human IL-18 ELISA kit (#E0147HU, BT Lab, Wuhan, China). In brief, supernatants were collected in sterile tubes and centrifuged at 2500 rpm for 10 min to remove any cell debris. For IL-1β detection, 100 μL of clear supernatant was collected and added to the pre-coated wells. For IL-18 detection, 40 μL of the supernatant was added to the plate and mixed with 10 μL of antibody, and 50 μL of streptavidin-HRP, in accordance with the manufacturer′s protocol. The plates were then incubated at 37 °C for an hour, the contents of both plates were aspirated and washed with the buffer supplied. After incubation with the detection antibody, a substrate solution was added to each well and incubated for 10 min. The reaction was stopped with stop solution and absorbance was recorded at 450 nm. The concentration of IL-1β and IL-18 was determined by calculating the appropriate standards that were provided.

### 4.8. NF-κB Reporter Assay

A luciferase-based NF-κB reporter system, kindly provided by Prof. Paulmurugan from Stanford University, School of Medicine, USA, was used to monitor NF-κB activity in cultured cells exposed to a hyperglycemic microenvironment. HEK293 cells expressing NF-κB-Fluc-Ubi-Rluc were seeded in a 24-well culture plate for this study. The cells were first exposed to the hyperglycemic microenvironment and co-treated with mangiferin (25 μM) at various time intervals, ranging from 30 min to 24 h. After treatment, the cells were washed, harvested, and lysed using cell lysis buffer (Promega, Madison, WI, USA). The cell supernatant was then collected, and luciferase activity was measured using a G20/20 tube luminometer (Promega, USA) after adding luciferase assay reagent provided with the kit (Promega, USA) [83].

### 4.9. DCFDA Staining and Fluorescent Microscopy

The total reactive oxygen species (ROS) in macrophages before and after MG treatment were estimated using a redox-sensitive fluorescent probe, 2’,7’-dichlorodihydrofluorescein diacetate (DCFDA) [84]. In brief, the cells were seeded on a glass coverslip and after exposure to HGM and MG, cells were washed with PBS, followed by the addition of 25 μM DCFDA (Invitrogen, Waltham, MA, USA) in fresh medium. The cells were incubated for about 30 min. Following this, the cells were again washed and incubated with 5 μg/mL DAPI (Invitrogen, USA) in fresh medium for an additional 10 min. After the incubation, the coverslips were retrieved, washed with PBS and mounted on a clean glass slide using fluoromount aqueous mounting medium (Sigma, USA). The mounted slides were then used to visualize ROS accumulation on HGM exposure and the effect of MG on it. The images were captured on a fluorescent microscope (Leica DM6, Nussloch, Germany) at wavelengths of 495/525 nm for DFCDA and 350/460 for DAPI.

### 4.10. In Vitro Cell Migration and Invasion Assay

Cell migration and invasion were performed using a Transwell Boyden chamber (Sigma, USA) to evaluate the effect of MG on macrophages [85]. The macrophages were exposed to MG and/or HGM as detailed above. The cells were then detached and counted, and a suspension containing 2 × 10^5^ cells in 100 μL of medium were seeded on to the upper part of the Transwell insert placed in a 24-well plate. The bottom of the lower chamber was added with fresh medium containing 10% FBS to facilitate the migration of cells. The setup incubated for 12 h. After incubation, the insert was removed and the spent medium was discarded. The insert was washed with PBS, then dried, and fixed with 600 μL of 70% ethanol for 10 min. After fixation, the insert was stained with 0.2% crystal violet stain for another 10 min, washed, dried, and then visualized under an inverted microscope to count migrated cells. Alternatively, to study the invasion potential, Matrigel basement extract (Corning, New York, NY, USA) was thawed, then 100 μL was added to the upper part of the chamber. This setup was ideal placed in the incubator for 30 min to allow the solidification of Matrigel, then 100 μL of cell suspension was added on top of the Matrigel to simulate invasion through this extracellular matrix.

### 4.11. Statistical Analysis

All data are represented as mean ± S.D. The graphs provided in this study were created with GraphPad Prism software (version 8.0). For statistical analysis, one-way ANOVA was performed in GraphPad Prism. Western blot densitometry was analyzed using ImageJ software (version 1.53). Mean fluorescence intensities for DCFDA and DAPI staining, as well as the number of migrating or invaded cells, were quantified using ImageJ software.

## 5. Conclusions

Our study demonstrates that mangiferin enhances Nrf2 activation and antioxidant defense in macrophages exposed to hyperglycemic stress, effectively mitigating inflammation and oxidative damage. By suppressing NF-κB and NLRP3 activation, mangiferin improves macrophage function and migration, thereby promoting wound healing. These findings highlight the potential of mangiferin as a promising therapeutic agent for diabetic wound management.

## Figures and Tables

**Figure 1 ijms-25-11197-f001:**
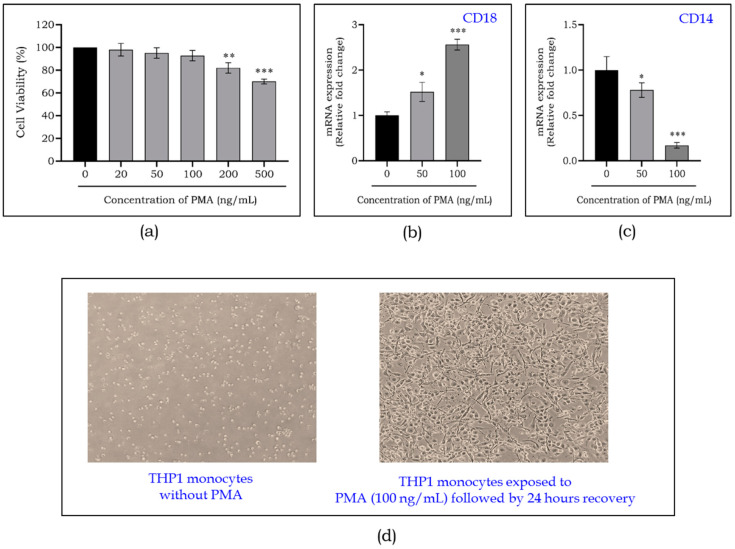
Viability of THP1 monocytes exposed to different concentrations of PMA for 24 h (**a**). Gene expression of CD18 (**b**) and CD14 (**c**) in PMA-exposed THP1 monocytes. Representative phase contrast inverted microscopy images of THP1 monocytes exposed to PMA for 24 h and after a recovery period of 24 h. Magnification 10× (**d**). Data are represented as mean ± S.D. * *p* < 0.05; ** *p* < 0.01; *** *p* < 0.001.

**Figure 2 ijms-25-11197-f002:**
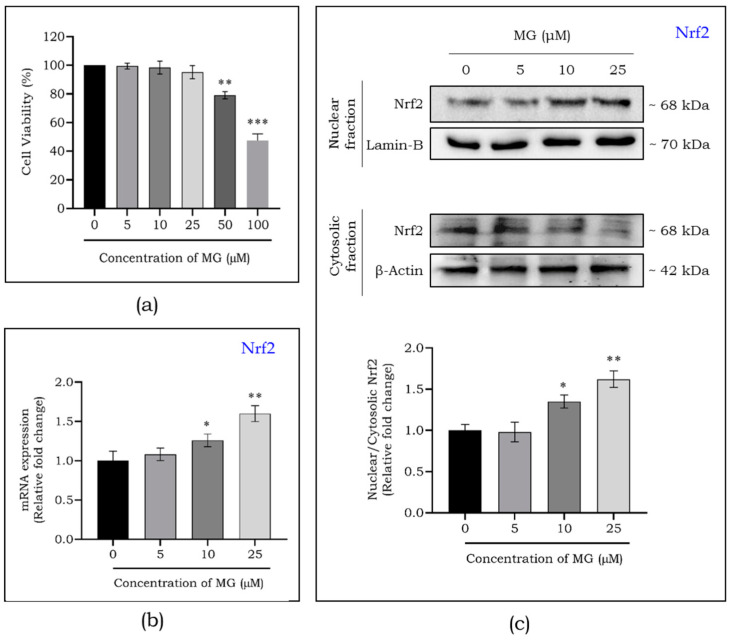
Dose-dependent cytotoxicity of mangiferin exposed to macrophages for a time period of 24 h (**a**). Nrf2 activation potential of mangiferin assessed by qPCR (**b**). Mangiferin-mediated Nrf2 translocation in macrophages (**c**). Data are represented as mean ± S.D. * *p* < 0.05; ** *p* < 0.01; *** *p* < 0.001; * significance compared to control. MG, mangiferin.

**Figure 3 ijms-25-11197-f003:**
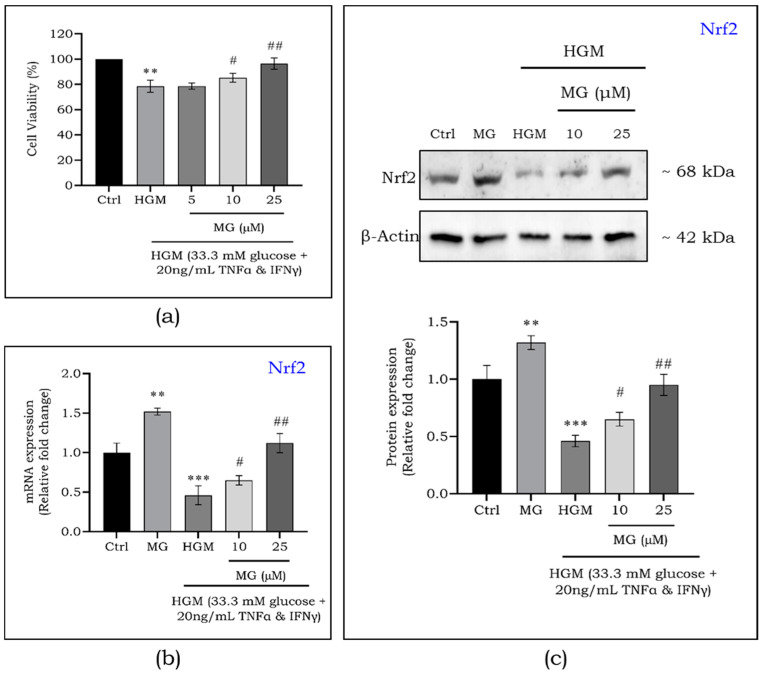
Protective effect of mangiferin over HGM in macrophages (**a**). Protein (**b**) and gene (**c**) expression of Nrf2in HGM-induced macrophages upon mangiferin treatment. Data are represented as mean ± S.D. *^#^ p* < 0.05; **^,##^ *p* < 0.01; *** *p* < 0.001; * significance compared to control; # significance compared to HG-induced macrophages. MG, mangiferin; HGM, hyperglycemic microenvironment.

**Figure 4 ijms-25-11197-f004:**
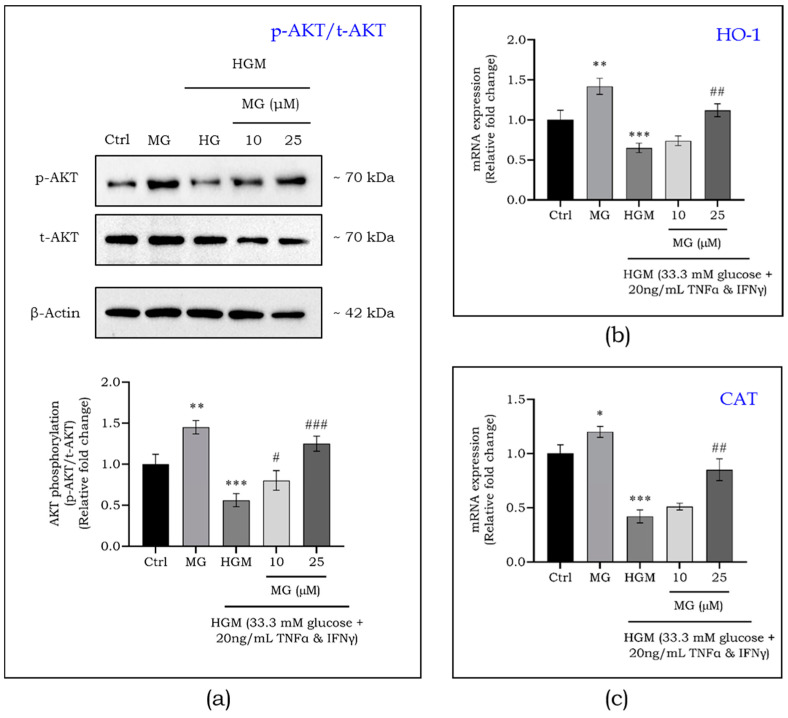
Mangiferin-induced phosphorylation of AKT (p-AKT/t-AKT) in macrophages exposed to HGM (**a**). Gene expression of Nrf2 downstream targets such as HO-1 (**b**) and CAT (**c**) in HGM-induced macrophages. Data are represented as mean ± S.D. *^,#^ *p* < 0.05; **^,##^ *p* < 0.01; ***^,###^ *p* < 0.001; * significance compared to control; # significance compared to HG-induced macrophages. MG, mangiferin; HGM, hyperglycemic microenvironment.

**Figure 5 ijms-25-11197-f005:**
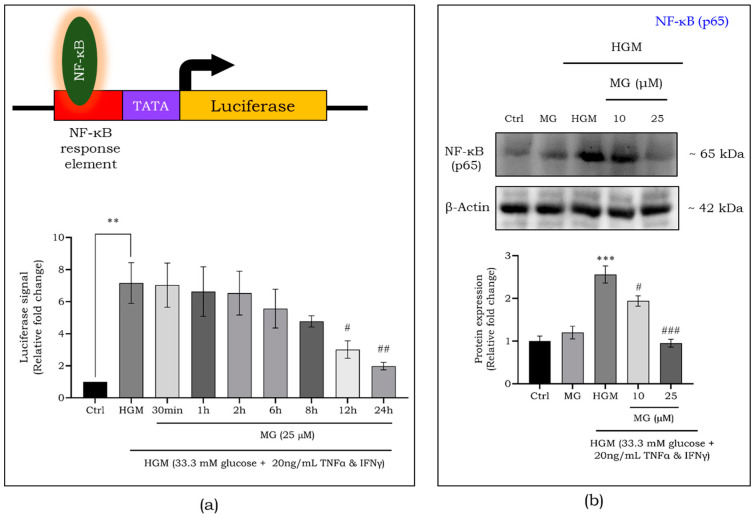
Effect of mangiferin on NF-κB activity as assessed by luciferase assay in HEK 293 cells stably expressing NF-κB luciferase (**a**), and NF-κB-p65 protein expression in HGM-induced macrophages (**b**). Data are represented as mean ± S.D. *^#^ p* < 0.05; **^,##^ *p* < 0.01; ***^,###^ *p* < 0.001; * significance compared to control; # significance compared to HGM-induced macrophages. MG, mangiferin; HGM, hyperglycemic microenvironment.

**Figure 6 ijms-25-11197-f006:**
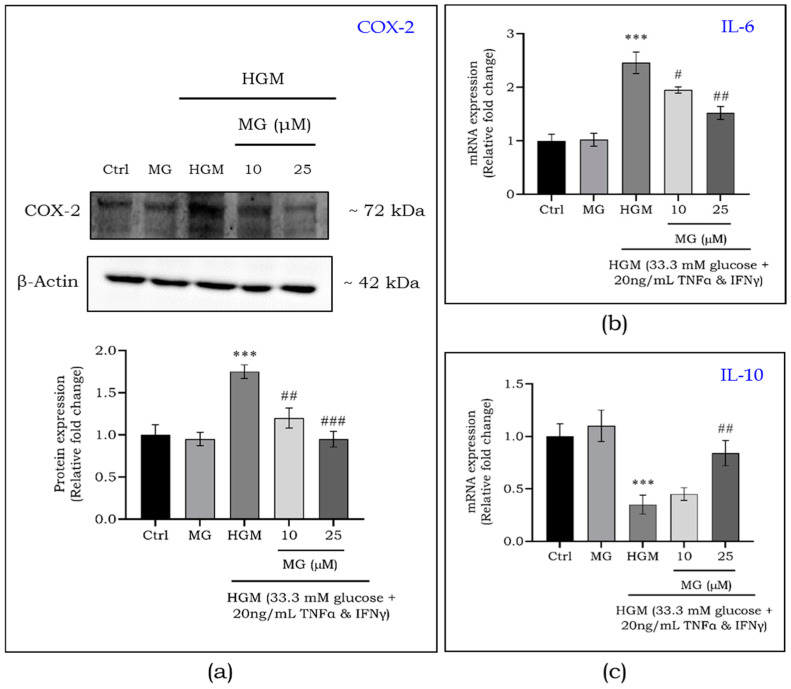
Effect of mangiferin on the protein expression of COX-2 (**a**) in HGM-induced macrophages. Gene expression of IL-6 (**b**) and IL-10 (**c**) in HGM-induced macrophages upon mangiferin treatment. Data are represented as mean ± S.D. ^#^ *p* < 0.05; ^##^ *p* < 0.01; ***^,###^ *p* < 0.001; * significance compared to control; # significance compared to HGM-induced macrophages. MG, mangiferin; HGM, hyperglycemic microenvironment.

**Figure 7 ijms-25-11197-f007:**
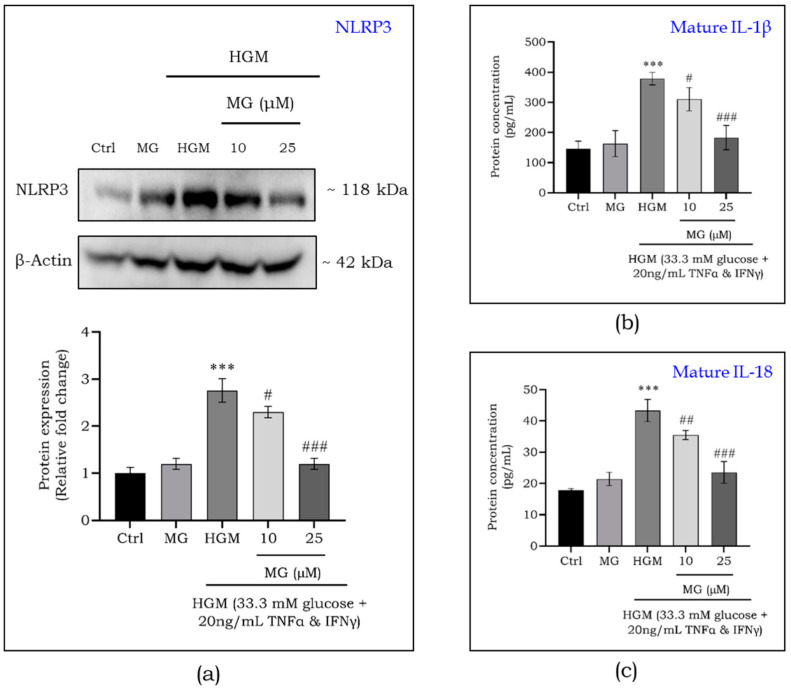
Effect of mangiferin on the protein expression of NLRP3 (**a**) in HGM-induced macrophages. Concentration of mature IL-1β (**b**) and IL-18 (**c**) in HGM-induced macrophages upon mangiferin treatment. Data are represented as mean ± S.D. ^#^
*p* < 0.05; ^##^ *p* < 0.01; ***^,###^ *p* < 0.001; * significance compared to control; # significance compared to HG-induced macrophages. MG, mangiferin; HGM, hyperglycemic microenvironment.

**Figure 8 ijms-25-11197-f008:**
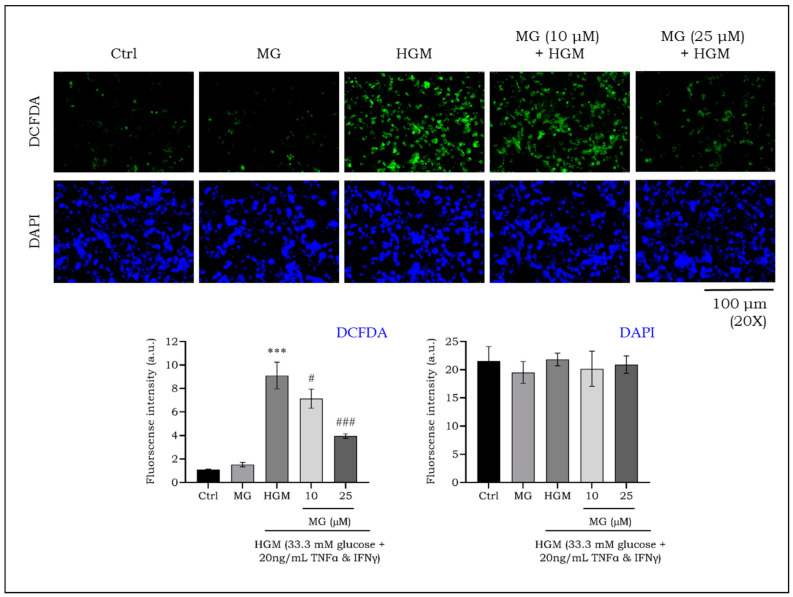
Effect of mangiferin on the formation of intracellular reactive oxygen species in HGM-induced macrophages. The cells were stained with DCFDA (green fluorescence) and counterstained with DAPI (blue). Images were captured on a fluorescent microscope. Bar graph was plotted by calculating the mean fluorescence. ^#^
*p* < 0.05; ***^,###^ *p* < 0.001; * significance compared to control; # significance compared to HG-induced macrophages. MG, mangiferin; HGM, hyperglycemic microenvironment.

**Figure 9 ijms-25-11197-f009:**
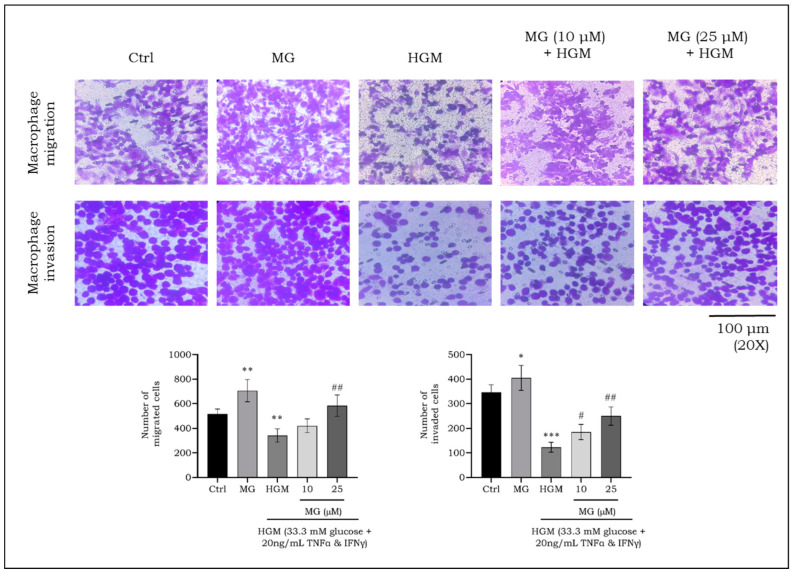
Effect of mangiferin on migration and invasion of HGM-induced macrophages. Images were captured using a phase contrast inverted microscope. The number of cells that migrated or invaded in a field view area was calculated and plotted as bar graph. ***^,#^ *p* < 0.05; **^,##^ *p* < 0.01; *** *p* < 0.001; * significance compared to control; # significance compared to HG-induced macrophages. MG, mangiferin; HGM, hyperglycemic microenvironment.

**Figure 10 ijms-25-11197-f010:**
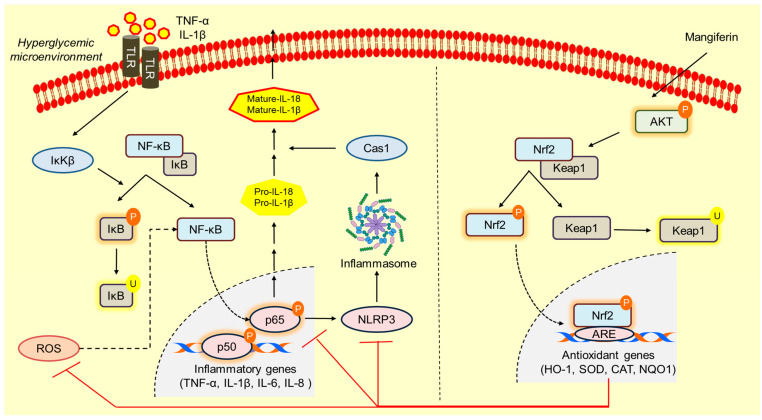
This schematic represents the sequence of events in macrophages exposed to hyperglycemia and the mitigation of hyperglycemia-induced stress by mangiferin. Hyperglycemia triggers the activation of inhibitor of kappa kinase (IκK), which promotes the dissociation of IκB from NF-κB. As a result, NF-κB translocates to the nucleus, where it transcribes inflammatory genes, including TNF-α, IL-1β, IL-6, and IL-8. Additionally, NF-κB enhances the synthesis of NLRP3 by binding to its response element, leading to inflammasome assembly. This, in turn, recruits caspase-1 (Cas1), which cleaves pro-IL-18 and pro-IL-1β into their mature, secreted forms. The secretion of mature IL-18 and IL-1β creates a pro-inflammatory environment, perpetuating inflammation in neighboring cells. Furthermore, secreted IL-1β can further activate IκK through toll-like receptors (TLRs), triggering a feed-forward inflammatory loop. Hyperglycemia also leads to the generation of reactive oxygen species (ROS), which further activates NF-κB, intensifying this cascade of inflammatory events. To counteract this inflammatory stress in macrophages, mangiferin activates AKT phosphorylation, which facilitates the dissociation of Keap1 from the Keap1-Nrf2 complex. This dissociation allows for the nuclear translocation of Nrf2, promoting the transcription of antioxidant genes such as HO-1, SOD, CAT, and NQO1. These antioxidant enzymes neutralize hyperglycemia-induced ROS accumulation and inhibit both NF-κB and NLRP3 activity, reducing inflammation. Therefore, mangiferin protects macrophages from the pro-inflammatory effects of hyperglycemia, promoting a shift toward an anti-inflammatory phenotype. This shift aids in tissue repair and helps prevent further tissue injury, offering a therapeutic approach for conditions characterized by chronic inflammation, such as diabetic wounds.

## Data Availability

Data available upon request.

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
