# Peer review of "Mangiferin Represses Inflammation in Macrophages Under a Hyperglycemic Environment Through Nrf2 Signaling"

_ijms, 2024, doi:10.3390/ijms252011197_

Round 1

Reviewer 1 Report

Comments and Suggestions for Authors

In this manuscript, Jayasuriya et al. report that Mangiferin suppresses hyperglycemia-induced macrophage activation through the induction of the Nrf2 signaling pathway. Overall, the authors address an interesting and relevant topic. The experiments are well designed, and the result is clearly written. The conclusions drawn are consistent with the observations presented. However, the manuscript's impact is somewhat limited by the fact that the activation of the Nrf2 signaling pathway by Mangiferin is already well documented, and the study offers little in terms of novel mechanistic insights into Mangiferin's mode of action. Additionally, a similar pathway has been previously reported by the same group in the context of preventing hyperglycemia-induced endothelial cell injury. Moreover, it remains unclear whether the in vitro findings can be translated to in vivo situations, which is a significant limitation.

Given the close relevance to diabetic pathology, the manuscript may still hold value for acceptance with minor revisions.

Minor comments:

Typographical errors should be corrected, such as:

Line 309: "bar graph" instead of "bar gram."

Line 187: "2 × 10^5" should be formatted correctly.

The authors should also clarify why control Mangiferin significantly increases macrophage migration and invasion (Fig. 7). 

Reviewer 2 Report

Comments and Suggestions for Authors

The manuscript titled "Mangiferin Represses Inflammation in Macrophages Under a Hyperglycemic Environment Through Nrf2 Signaling" offers valuable insights into the anti-inflammatory effects of mangiferin, particularly in the context of diabetic wounds. The study effectively demonstrates how mangiferin modulates the Nrf2 pathway to address macrophage inflammation and oxidative stress. However, a few areas could benefit from further refinement to enhance the overall understanding. While the manuscript highlights key pathways, there is limited exploration of other relevant pathways, such as MAPK or PI3K/Akt, which could provide additional support for the findings. Additionally, a more detailed discussion of potential synergies between mangiferin and other compounds and comparison with standard anti-inflammatory drugs would strengthen the manuscript. Furthermore, the reliance on some older references, particularly in the Introduction, could be updated to improve the manuscript's relevance.

I have attached my specific questions and suggestions for your revision. Addressing these points will improve the manuscript's clarity, scientific rigor, and overall impact. My comments are outlined below, section by section.

Round 2

Reviewer 2 Report

Comments and Suggestions for Authors

Thank you for addressing the majority of my concerns. While I appreciate the revisions made, I believe a few points, particularly regarding the control experiments, are still not fully satisfactory. However, since the major concerns have been appropriately addressed, I am happy to recommend accepting the manuscript in its revised form.